# High-risk human papillomavirus prevalence in self-collected cervicovaginal specimens from human immunodeficiency virus (HIV)-negative women and women living with HIV living in Botswana

**Philip E. Castle[1], John E. Varallo[2]\*, Margaret Mary Bertram[2], Bakgaki Ratshaa[3], Moses Kitheka[3], Kereng Rammipi[4]**

**1** Department of Epidemiology and Population Health, Albert Einstein College of Medicine, Bronx, NY, United States of America, **2** Jhpiego USA, Baltimore, MD, United States of America, **3** Jhpiego Botswana—Gaborone, Botswana, **4** Botswana Ministry of Health and Wellness, Gaborone, Botswana

\* John.Varallo@jhpiego.org

**Data Availability Statement:** All relevant data are within the paper.

## Abstract

### Background

The prevalence of high-risk human papillomavirus (hrHPV) is poorly described overall and in women living with HIV (WLWH) and HIV-negative women living in Botswana, a high HIV and cervical cancer-burden country. We conducted a pilot study of self-collection and high-risk HPV testing for cervical screening, from which data on HPV prevalence was available.

### Methods

From five health facilities in the Kweneng East District, 1,022 women aged 30–49 years were enrolled to self-collect their cervicovaginal specimen for hrHPV testing by the Xpert HPV Test (Cepheid, Sunnyvale, CA, USA). Crude and age group-adjusted hrHPV prevalence by HIV status were calculated, and the relationship of hrHPV risk groups HPV16>HPV18/45>other hrHPV types) to the presence and severity of visible lesions.

### Results

Of the 1,022 women enrolled, 1,019 (99.7%), 570 WLWH and 449 HIV-negative women, had hrHPV testing results. Crude hrHPV prevalences were 25.2% (95%CI = 21.2–29.4%) for HIV-negative women and 40.4% (95%CI = 36.3–44.5%) for WLWH. Age group-adjusted hrHPV prevalences were 23.7% (95%CI = 19.9–27.9%) for HIV-negative women and 41.3% (95%CI = 37.2–45.4%) for WLWH. Age group-adjusted prevalences of HPV16 (p<0.001), HPV18/45 (p<0.001), HPV31/33/35/52/58 (p<0.001), and HPV39/56/66/68 (p = 0.011) were greater among WLWH than HIV-negative women. Riskier hrHPV groups were more likely to have visible abnormalities ($p_{trend}$ = 0.004) and visible abnormalities not eligible for cryotherapy ($p_{trend}$ = 0.030).

**Funding:** This study was a Jhpiego (https://www.jhpiego.org/) internally funded project to Moses Kitheka. The funder had no role in study design, data collection and analysis, decision to publish, or preparation of the manuscript.

**Competing interests:** Philip E. Castle has received discounted or free HPV tests and assays for research purposes from Roche, Bectin Dickinson, Cepheid, and Arbor Vita Corporation. This does not alter our adherence to PLOS ONE policies on sharing data and materials.

## Conclusions

hrHPV infection was common among all women in the study living in Botswana, to a greater extent in WLWH than their HIV-negative counterparts. Strategies to triage hrHPV-positive women will be needed to avoid over-treating many women with benign hrHPV infections.

## Introduction

Testing for high-risk human papillomavirus (hrHPV), cause of virtually all cervical cancer, is increasingly the preferred method for screening to prevent cervical cancer in mid-adult women living in high-income countries (HIC) and low- and middle-income countries (LMICS) [1–3]. One of the important advantages of hrHPV testing-based screening over cytology-based screening is that the former empowers women by allowing them to self-collect their own cervicovaginal specimen rather than needing to have a pelvic exam to get a provider-collected specimen and permits them to collect their own specimen in privacy, as well as elsewhere than at the clinic. Importantly, there is strong evidence that the use of self-collected cervicovaginal specimen with hrHPV testing is as accurate as using a provider-collected cervical specimen, increases participation in screening, and self-collection at home is preferred over clinic-based screening [4–7].

To examine the feasibility of introducing HPV testing of self-collected vaginal samples and a hrHPV screen-and-treat algorithm in Botswana, a high HIV and cervical-cancer burden country where cervical cancer is the leading cause of cancer-related deaths among women [8], we conducted a pilot study among women attending five public health facilities (one district hospital and four health clinics and surrounding communities) in the Kweneng East District in Botswana in a convenience sample of approximately 1,000 women, enriched for women living with HIV (WLWH). Women who were seeking sexual and reproductive health (SRH) services were asked to self-collect their specimen, which was tested for hrHPV, and hrHPV-positive women underwent visual assessment and treatment according to World Health Organization (WHO) recommendations [1;9]. Here, we report the hrHPV testing results, using a WHO prequalified hrHPV test [10], in relationship to basic demographic data and HIV status (positive vs. negative) as well as what treatment these women underwent.

## Methods

### Study ethics approval

Before commencing the study, the Johns Hopkins School of Public Health (JHSPH) Institutional Review Board approved the study (reference IRB00007974).

A convenience sample of 1,022 women aged 30–49 years living in Kweneng East District in Botswana, which serves approximately 46,000 women in this age group, was recruited to participate in a pilot demonstration project on self-collection-based HPV screening. Women were recruited in equal (20%) proportions, between 203–206 participants from five health facilities, designated as A, B, C, D, and E, to insure anonymity. Dates for recruitment were October 2017 –March 2018. All five health facilities conducted recruitment at their respective facilities, while two health facilities (B and E) also conducted recruitment in the community during health outreach activities. In these two facilities, recruitment was planned to be half and half from the health facility and in the community each serves through existing integrated community health outreaches, but 70% were recruited from the community health outreach

for one facility (B). Women at the clinics were recruited after coming to the facility for cervical cancer screening following contact with a research nurse and from various health service points at the clinic, such as family planning, outpatient, and HIV care. Women attending the latter were specifically targeted to enrich the study sample for WLHW.

Eligible women were recruited by the research nurses who conducted group education on HPV and cervical cancer, the ways to prevent it, and described the study. Eligible and interested women then met individually with the research nurse to confirm eligibility and interest. Eligible women who remained interested in participating in the study were then consented and enrolled. The research nurses obtained written informed consent of the study participants by providing a written copy of the consent form, in Setswana, to the participant and reading the form to them, confirming understanding of all aspects of the study, including volunteer participation. The research nurse then printed the client's name and obtained her signature and date of the informed consent. Consented women were then instructed on how to self-collect their specimen. For community recruitment, the study used the existing community health campaign mechanism to integrate cervical cancer screening using HPV self-collection into these community outreach events.

Inclusion/exclusion criteria were: 1) aged 30 to 49 years; 2) not screened recently/never screened before, defined as self-report of no prior history of cervical cancer screening, prior screening but result unknown and no treatment, or screening occurred more than 5 years ago for HIV negative women or 3 years ago for WLHW; 3) HIV status known (HIV positive result, or documented HIV negative result is less than 12 months ago); 4) no history of prior abnormal screening or treatment/procedure on her cervix due to abnormal screening; 5) no history of cervical cancer; 6) not currently pregnant and not less than 6 weeks postpartum; 7) an intact uterus/no prior hysterectomy with complete removal of the cervix; 8) accesses health services in Kweneng East District study catchment area; and 9) able and willing to provide consent.

The research nurse collected basic demographic and contact information, including telephone number(s) to allow for follow up with results. The Senior Research Nurse then distributed HPV self-collection kits which included a sampling brush (using Viba-Brush® (Rovers Medical Devices BV, Oss, the Netherlands) and a PreservCyt solution vial (Hologic, Bedford, MA, USA) to the woman, and she was instructed on proper self-collection technique. The woman then went to a designated private area to self-collect the cervicovaginal specimen, rinsed the collection device in the PreservCyt vial to elute the cells, capped the vial, and returned the vial to a research nurse.

The self-collected specimens were tested using the Xpert HPV Test, a qualitative, real-time PCR assay for the detection of hrHPV DNA per the manufacturer's instructions [10]. The Xpert HPV assay includes simultaneous detection of 14 hrHPV types, hydroxymethylbilane synthase (HMBS) and an internal Probe Check Control. The 14 targeted hrHPV genotypes (HPV16, 18, 31, 33, 35, 39, 45, 51, 52, 56, 58, 59, 66, and 68) are detected in five fluorescent channels: 1) HPV16, 2) HPV18 and 45 (HPV18/45), 3) HPV31, 33, 35, 52, and 58 (HPV31/33/35/52/58), 4) HPV51 and HPV59 (HPV51/59), and 5) HPV39, 56, 66, and 68 (HPV39/56/66/68).

Women who tested hrHPV positive were asked to undergo further evaluation with a pelvic exam, at which time dilute acetic acid was placed on the cervix to perform visual assessment for treatment (VAT), i.e., to evaluate any cervical abnormalities that became white after dilute acetic acid was applied to the cervix ("acetowhite"), and decide the recommended treatment according to WHO guidelines [9;11]. Nurses, midwives, and doctors, previously trained in visual inspection with acetic acid (VIA) and cryotherapy, performed VAT and triaged every HPV-positive client to determine treatment method. Abnormalities that covered less than three-quarters of the cervix, were completely visible i.e., did not go into the endocervical canal,

and were not suspicious of cancer were deemed eligible for ablation and treated by cryotherapy. Those that covered three-quarters or more of the cervix and/or went into the endocervical canal were deemed ablation ineligible and were referred for loop electrosurgical excision procedure (LEEP). Those with suspected cancer were referred to colposcopy and biopsy and based on those results were then referred for care.

**Analysis.** Age was categorized as 30–34, 35–39, 40–44, and 45–50 years. Crude and age group-adjusted prevalence, with 95% confidence intervals (95%CI) of any hrHPV and the individual hrHPV groups defined by the Xpert HPV Test channels, were calculated. Differences in crude and age group-adjusted prevalence between WLWH and HIV-negative women were tested for statistical significance (p<0.05) using Fisher's exact and Wald chi-square tests, respectively. Unadjusted (crude) odds ratios (OR) and age group-adjusted ORs (aOR), with 95%CI, were calculated using logistic regression as a measure of the association of HIV status with hrHPV prevalence.

Differences in age between WLWH and HIV-negative women were tested for statistical significance using the Kruskal-Wallis Test. A linear regression model was used to predict the age-specific hrHPV for WLWH and HIV.

Crude and age group-adjusted hrHPV prevalence was compared across clinical sites and tested for differences using a Pearson chi-square and Wald chi-square tests, respectively. Differences in hrHPV prevalence between WLWH and HIV-negative women for a given site, or between community vs. facility recruitment by site and HIV status, were tested for statistical significance using a Fisher's exact test. Logistic regression models were used to calculate OR and 95%CI as a measure of association of age group, facility, and HIV status with hrHPV prevalence.

The outcome of VAT (no visible abnormality, visible abnormality and cryotherapy eligible, visible abnormality and cryotherapy ineligible, or suspected cancer) and hrHPV results, categorized hierarchically according to their cancer risk (HPV16 positive, versus HPV16 negative and positive for HPV18/45, versus HPV16 and 18/45 negative and positive for other hrHPV types), were compared. VAT outcomes and hrHPV risk group were tested for statistical significance using a test for trend [12].

## Results

The study recruited 1022 eligible women, 571 WLWH and 451 HIV-negative women; 1019 women had hrHPV results, 570 WLWH and 449 HIV-negative women. Although the age eligibility for the study was restricted to 30–49 years, the HIV-negative women (mean = 37.5, median = 36, and IQR = 33–42 years) enrolled in the study were significantly younger than the WLWH (mean = 39.4, median = 39, and IQR = 35–43 years) (p<0.001).

Crude hrHPV prevalences were 25.2% (95%CI = 21.2–29.4) for HIV-negative women and 40.4% (95%CI = 36.3–44.5) for WLWH (**Table 1**). hrHPV prevalence decreased with increasing age for both HIV-negative women and WLWH (p<0.001 for both) (**Fig 1**). Because of the difference in age between the WLWH and HIV-negative women and the relationship of age with hrHPV prevalence, we also adjusted the hrHPV prevalence estimates for age. Age group-adjusted hrHPV prevalences were 23.7% (95%CI = 19.9–27.9%) for HIV-negative women and 41.3% (95%CI = 37.2–45.4%) for WLWH. The age group-adjusted OR for the association of being WLWH (vs. HIV-negative women) with hrHPV detection was 2.3 (95% CI = 1.7–3.0).

Prevalences of most hrHPV groups, as determined by the Xpert HPV Test, also differed by HIV status (**Table 1**). The age group-adjusted prevalence was lower for HIV-negative women than WLWH for HPV16 (2.8% vs. 8.3%, respectively, p<0.001), HPV18/45 (3.7% vs. 10.9%,

**Table 1. Prevalence of high-risk human papillomavirus (hrHPV) overall and individual hrHPV groups, as detected by the Xpert HPV Test, for human immunodeficiency virus (HIV)-negative women (n = 449) and women living with HIV (WLWH) (n = 570).**

| HPV Test Result | HIV-Negative Women | | | WLWH | | | | | | |
|---|---|---|---|---|---|---|---|---|---|---|
| | $N_{hrHPV+}$ | Crude (95% CI) | Age Adjusted (95% CI) | $N_{hrHPV+}$ | Crude (95% CI) | Age Adjusted (95% CI) | P* | OR* (95%CI) | p** | OR** (95%CI) |
| Any hrHPV[†] | 113 | 25.2 (21.2–29.4) | 23.7 (19.9–27.9) | 230 | 40.4 (36.3–44.5) | 41.3 (37.2–45.4) | <0.001 | 2.0 (1.5–2.6) | <0.001 | 2.3 (1.7–3.0) |
| Channel 1 HPV16 | 14 | 3.1 (1.7–5.1) | 2.8 (1.6–4.7) | 47 | 8.2 (6.1–10.8) | 8.3 (6.3–10.9) | 0.001 | 2.8 (1.5–5.1) | <0.001 | 3.2 (1.7–5.9) |
| Channel 2: HPV18/45 | 18 | 4.0 (2.4–6.4) | 3.7 (2.3–5.9) | 61 | 10.7 (8.3–13.5) | 10.9 (8.5–13.7) | <0.001 | 2.9 (1.7–4.9) | <0.001 | 3.1 (1.8–5.4) |
| Channel 3: HPV31/33/35/52/58 | 48 | 10.7 (8.0–13.9) | 10.1 (7.7–13.3) | 121 | 21.2 (17.9–24.8) | 21.6 (18.4–25.2) | <0.001 | 2.3 (1.6–3.2) | <0.001 | 2.4 (1.7–3.5) |
| Channel 4: HPV51/59 | 23 | 5.1 (3.3–7.6) | 5.0 (3.3–7.5) | 30 | 5.3 (3.6–7.4) | 5.3 (3.7–7.5) | 1 | 1.0 (0.6–1.8) | 0.811 | 1.1 (0.6–1.9) |
| Channel 5: HPV39/56/66/68 | 29 | 6.5 (4.4–9.1) | 6.2 (4.3–8.8) | 62 | 10.9 (8.4–13.7) | 10.7 (8.4–13.5) | 0.02 | 1.7 (1.1–2.7) | 0.011 | 1.8 (1.1–2.9) |
| Channel 1, 2, or 3[†]: HPV16/18/31/33/35/45/52/58 | 75 | 16.7 | 15.6 (12.5–19.2) | 189 | 33.2 (29.3–37.2) | 33.9 (30.0–37.9) | <0.001 | 2.5 (1.8–3.5) | <0.001 | 2.8 (2.0–3.8) |

*crude hrHPV prevalence in WLWH vs. HIV-negative women

**age-adjusted hrHPV prevalence in WLWH vs. HIV-negative women

[†]Because some women were positive for more than one of the channels, the number of positives does not equal the sum of the individual channels

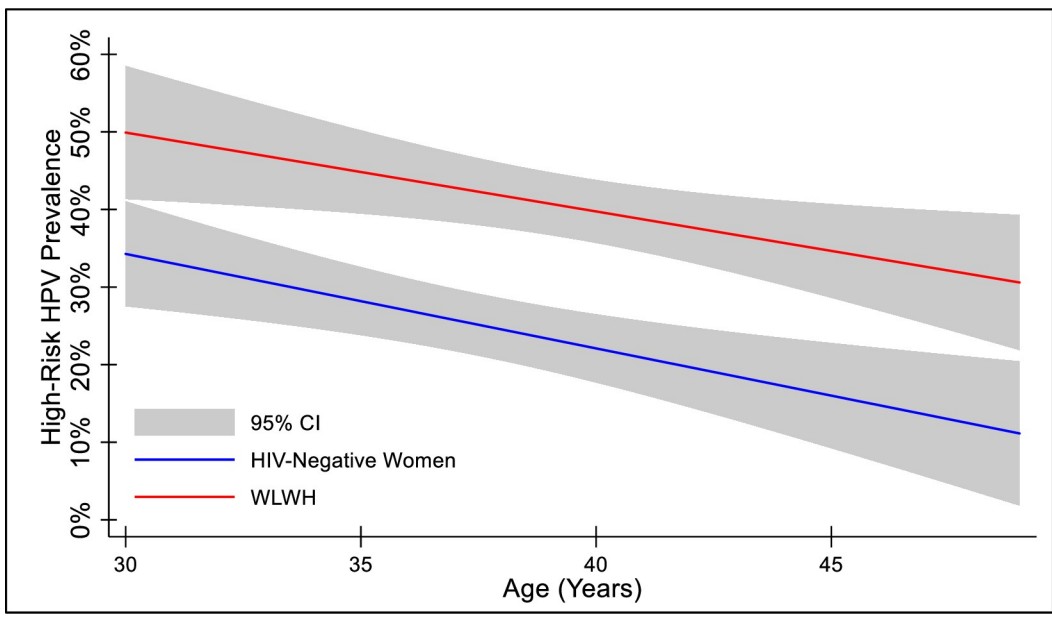

**Fig 1. Prediction of high-risk human papillomavirus (HPV) prevalence, stratified by HIV status, by age.** Abbreviation: HIV, human immunodeficiency virus; WLWH, women living with HIV.

**Table 2. High-risk human papillomavirus (hrHPV) prevalence by health facility and human immunodeficiency virus (HIV) status.** Abbreviations: WLWH, women living with HIV.

| Study Health Facility | HIV-Negative Women | | | | WLWH | | | | | |
|---|---|---|---|---|---|---|---|---|---|---|
| | N | N(hrHPV+) | Crude | Age Adjusted | N | N(hrHPV+) | Crude | Age Adjusted | p* | p** |
| A | 84 | 30 | 35.7 | 34.3 | 117 | 49 | 41.9 | 42.9 | 0.385 | 0.163 |
| B | 92 | 15 | 16.3 | 15.7 | 113 | 41 | 36.3 | 34.1 | 0.002 | 0.001 |
| C | 72 | 14 | 19.4 | 19.6 | 131 | 60 | 45.8 | 46.4 | <0.001 | <0.001 |
| D | 93 | 28 | 30.1 | 29.0 | 111 | 38 | 34.2 | 34.1 | 0.551 | 0.475 |
| E | 108 | 26 | 24.1 | 23.7 | 98 | 42 | 42.9 | 42.7 | 0.005 | 0.002 |
| Total | 449 | 113 | 25.2 | 23.7 | 570 | 230 | 40.4 | 41.3 | <0.001 | <0.001 |
| p† | | | 0.024 | 0.034 | | | 0.345 | 0.196 | | |

*difference in crude hrHPV prevalence in WLWH vs. HIV-negative women

**difference in age-adjusted hrHPV prevalence in WLWH vs. HIV-negative women

†difference in hrHPV prevalence across facilities

respectively, p<0.001), HPV31/33/35/52/58 (10.1% vs. 21.6%, respectively, p<0.001) and HPV39/56/66/68 (6.2 vs. 10.7%, respectively, p = 0.011). However, there was no significant difference in the age group-adjusted prevalence of HPV51/59 between HIV-negative women and WLWH (5.0 vs. 5.3%, respectively, p = 0.811). WLWH were twice as likely as HIV-negative women to be positive for multiple HPV groups (30% vs. 15.9%, respectively, p = 0.005).

We examined the hrHPV prevalence by clinical site (Table 2). Age group-adjusted hrHPV prevalences varied significantly by clinical site among the HIV-negative women (range = 15.7% to 34.3%, p = 0.034). By comparison, the hrHPV prevalence was less variable by clinical site among WLWH (range = 34.1% to 46.4%, p = 0.196). Surprisingly, there were two health facilities, A and D, for which the hrHPV prevalence did not differ significantly by HIV status.

Interestingly, the hrHPV prevalence differed between those recruited from the community or at the health facility for two (B and E) health facilities but only for HIV-negative women (Table 3). At one health facility (B), the hrHPV prevalence was 30.4% in those recruited at the facility vs. 11.6% in those recruited from community (p = 0.050) among HIV-negative women. The hrHPV prevalence did not differ significantly between WLWH recruited at the facility or from community (35.1% vs. 36.8%, respectively, p = 1.000). In contrast, at another health facility (E), the hrHPV prevalence was 15.8% in those recruited at the facility vs. 33.3% in those recruited from community (p = 0.043) among HIV-negative women. Again, the hrHPV prevalence did not differ significantly between WLWH recruited at the facility or from community (39.1% vs. 46.2%, respectively, p = 0.543).

In a logistic regression model, HIV status, health facility, and age group were all independent predictors of hrHPV prevalence (Table 4). Being HIV positive was strongly associated with having a hrHPV infection (aOR = 2.35, 95%CI = 1.76–3.14). Older women were less likely

**Table 3. Prevalence of high-risk human papillomavirus (hrHPV) by HIV status (HIV-negative women or women living with HIV (WLWH)) and recruitment strategy (facility or community).**

| Study Health Facility | HIV-Negative | | | | | WLWH | | | | |
|---|---|---|---|---|---|---|---|---|---|---|
| | Facility | | Community | | p | Facility | | Community | | p |
| | N | %hrHPV+ | N | %hrHPV+ | | N | %hrHPV+ | N | %hrHPV+ | |
| B | 23 | 30.4 | 69 | 11.6 | 0.050 | 37 | 35.1 | 76 | 36.8 | 1.00 |
| E | 57 | 15.8 | 51 | 33.3 | 0.043 | 46 | 39.1 | 52 | 46.2 | 0.543 |

**Table 4. Crude odds ratio (OR) and adjusted odds ratios (aOR), with 95% confidence intervals (95%CI), as measures of association of human immunodeficiency virus (HIV) status, health facility, and age-group with prevalent high-risk human papillomavirus (hrHPV) in women living in Botswana.**

| Variable | | N (%) | OR | aOR (95%CI) |
|---|---|---|---|---|
| HIV Status | | | | |
| | Negative (ref) | 449 (44.06%) | 1.0 | 1.0 |
| | Positive | 570 (55.94%) | **2.01 (1.53–2.64)** | **2.35 (1.76–3.14)** |
| Facility | | | | |
| | A (ref) | 201 (19.73%) | 1.0 | 1.0 |
| | B | 205 (20.12%) | **0.58 (0.38–0.88)** | **0.53 (0.35–0.82)** |
| | C | 203 (19.92%) | 0.89 (0.59–1.32) | 0.84 (0.56–1.27) |
| | D | 204 (20.02%) | 0.74 (0.49–1.11) | 0.71 (0.47–1.08) |
| | E | 206 (20.22% | 0.76 (0.51–1.14) | 0.79 (0.52–1.20) |
| Age Group (Years) | | | | |
| | 30–34 (ref) | 271 (26.59%) | 1.0 | 1.0 |
| | 35–39 | 305 (29.93%) | **0.63 (0.45–0.89)** | **0.50 (0.35–0.72)** |
| | 40–44 | 277 (27.18%) | 0.79 (0.56–1.11) | **0.60 (0.42–0.86)** |
| | 45–49 | 166 (16.29%) | **0.42 (0.27–0.65)** | **0.31 (0.20–0.49)** |

to have hrHPV infection than younger women, with women aged 45–49 years the least likely to have a hrHPV infection (aOR = 0.31, 95%CI = 0.20–0.49) compared to women aged 30–34 years.

Finally, among hrHPV-positive women, the relationship of hierarchical hrHPV risk groups (HPV16>HPV18/45>other hrHPV types) and the visual status of the cervix (no visible lesion, visible abnormalities that was eligible for ablation, visible lesion that was ineligible for ablation, and suspected cancer) was examined (**Table 5**). Riskier hrHPV groups were more likely to have visible abnormalities (42% for HPV16, 26% for HPV18/45, and 22% for other high-risk HPV; $p_{trend}$ = 0.004) and visible abnormalities not eligible for cryotherapy (19% for HPV16, 12% for HPV18/45, and 8% for other high-risk HPV; $p_{trend}$ = 0.030). Riskier hrHPV groups were more likely to have visible abnormalities (50% for HPV16, 18% for HPV18/45, and 17%

**Table 5. The relationship of high-risk human papillomavirus (hrHPV) and hrHPV risk groups and results of visual assessment for treatment.**

| HPV Category | | Total | A. No Visible Abnormality: Ablation Eligible | B. Visible Abnormality: Ablation Eligible | C. Visible Abnormality: Ablation Ineligible* | D. Cancer Suspected: Ablation Ineligible* | Visible Lesions** |
|---|---|---|---|---|---|---|---|
| hrHPV Positive | N | 328 | 240 | 52 | 32 | 4 | 88 |
| | % Row | 100% | 73% | 16% | 10% | 1% | 27% |
| HPV 16 Positive | N | 61 | 34 | 14 | 10 | 1 | 25 |
| | % Row | 100% | 56% | 23% | 16% | 2% | 42% |
| HPV18/45 Positive; HPV16 Negative | N | 68 | 50 | 10 | 7 | 1 | 18 |
| | % Row | 100% | 74% | 15% | 10% | 1% | 26% |
| Other hrHPV Positive; HPV16 and 18/45 Negative | N | 201 | 156 | 28 | 15 | 2 | 45 |
| | % Row | 100% | 78% | 14% | 7% | 1% | 22% |

*$p_{trend}$ = 0.030 for HPV groups vs. ablation ineligible ((C + D)/Total)

**$p_{trend}$ = 0.004 for HPV groups vs. visible lesions ((B + C + D)/Total)

**Table 6. The relationship of high-risk human papillomavirus (hrHPV) and hrHPV risk groups and the presence of visible (acetowhite) cervical abnormalities, stratified on HIV status.**

| HPV Category | HIV-Negative Women* | | | WLWH** | | | p† |
|---|---|---|---|---|---|---|---|
| | N | N (Visible Lesion) | %Visible Abnormality | N | N (Visible Lesion) | %Visible Abnormality | |
| hrHPV Positive | 109 | 23 | 21% | 219 | 65 | 30% | 0.113 |
| HPV 16 Positive | 14 | 7 | 50% | 45 | 18 | 40% | 0.549 |
| HPV18/45 Positive; HPV16 Negative | 17 | 3 | 18% | 51 | 15 | 29% | 0.527 |
| Other hrHPV Positive; HPV16 and 18/45 Negative | 78 | 13 | 17% | 123 | 32 | 26% | 0.164 |

* $p_{trend}$ = 0.013

** $p_{trend}$ = 0.091

† HIV-negative women vs. WLWH

for other high-risk HPV; $p_{trend}$ = 0.013) among HIV-negative women (**Table 6**). Riskier hrHPV groups were marginally more likely to have visible abnormalities among WLWH (40% for HPV16, 29% for HPV18/45, and 26% for other high-risk HPV; $p_{trend}$ = 0.091) (**Table 6**).

## Discussion

In this pilot study of self-collection-based HPV screening of approximately one-thousand women living in Botswana, we made the following observations: 1) age-adjusted hrHPV prevalence was almost 2-fold higher in WLWH than HIV-negative women; notably, the hrHPV prevalence in HIV-negative women was high and quite variable between health facilities; 2) age, health facility, and HIV status were all predictors of prevalence hrHPV infection; and 3) riskier hrHPV groups were more likely to have visible cervical abnormalities and notably ablation-ineligible visible cervical abnormalities; approximately one-half and one-fifth of prevalent HPV16 had a visible cervical abnormality and ablation-eligible visible cervical abnormality, respectively.

There are few data published on the prevalence of cervical/cervicovaginal hrHPV in women living in Botswana. Luckett *et al*. [13] reported the hrHPV prevalence, as detected by Xpert on provider-collected cervical specimens, to be 29% in 300 WLWH with a median and interquartile range of age of 46 and 42–52 years, respectively. A sub-study (n = 103) in this same population of self-collection and hrHPV testing found 27% hrHPV prevalence in self-collected specimens [14]. Macleod *et al*. [15] reported the hrHPV prevalence, as detected by Linear Array (Roche, Pleasanton, CA, USA) on provider-collected cervical specimens, to be 25% in 139 WLWH with a median and interquartile range of age of 46 and 42–52 years, respectively. We found no published reports of the hrHPV prevalence in HIV-negative women.

The HPV prevalence in WLWH reported in this study was lower than observed in some populations [16–22], comparable to some populations [18;23–26], and higher than in other populations [18;27] living in SSA. We observed, as seen in other studies, that HIV status [24;27] was an independent predictors of HPV prevalence in WLWH. However, the relative prevalence of hrHPV in WLWH vs. HIV-negative women, less than two-fold, was rather low compared to other studies that reported 2.5-fold or greater relative prevalence [20–22;24;28;29]. This was in part due to the relatively high hrHPV prevalence in HIV-negative women, which was driven by high hrHPV prevalence (>30%) in HIV-negative women recruited at certain health facilities. Age across sites was only marginally different (p = 0.055) and therefore probably does not explain the heterogeneity in hrHPV prevalence between

them. Alternatively, higher-risk HIV-negative women were more likely to be recruited into and/or volunteer to participate in the study at some sites.

These data highlight the need for effective triage strategies for hrHPV-positive women, especially in WLWH populations, to increase the specificity and reduce the unnecessary treatment of benign hrHPV infections. Visual inspection after acetic acid (VIA) has been recommended as a triage of hrHPV [1] and would have cut down the number of women treated by ~73%. However, there are a number of limitations of using VIA as a triage test for an HPV-positive test including only moderate clinical performance, notably reduced sensitivity for high-grade cervical abnormalities [13;30;31]. A promising new strategy is the use of a deep learning-based automated visual evaluation tool that may provide real-time image analysis to identify sensitively and specifically those with high-grade cervical abnormalities and early cervical cancer [32].

HPV genotyping could also be used as the triage. Triage with HPV16, the HPV type responsible for 50–60% of cervical cancers [33], would reduce treatment by 82%. Triage with HPV16 and HPV18/45, HPV types responsible for ~75% of the cervical cancer [33], would reduce treatment by 62%. Alternatively, limiting the definition of a positive hrHPV test to the 8 hrHPV types (HPV16, HPV 18/45, and HPV31/33/35/52/58) that cause ~90% of cervical cancers [33] would reduce hrHPV positivity by approximately 34% in HIV-negative women and 18% in WLWH.

We found the HPV type, notably HPV16, was related to the appearance of the lesion. A study of women diagnosed with CIN2+ living in China found that the presence of HPV16 was associated with the presence of acetowhite lesions [34]. A study of unscreened women living Papua New Guinea found that HPV16 was also associated with VIA positivity [35]. In contrast, another study in The Netherlands and Spain did not find that any relationship of colposcopic appearance of a lesion [36]. We hypothesize that HPV16 infections cause the most obvious and severe appearing lesions, perhaps not surprisingly since HPV16 is the most carcinogenic HPV genotype [33;37], and therefore the most likely abnormalities to be detected and treated in a well-screened population, like those of the Netherlands and Spain.

We acknowledge several limitations of our study. First, because this was a convenience sample of women, the true hrHPV prevalence in the whole population, as well as in WLWH and HIV-negative women cannot be truly estimated. The heterogeneity of hrHPV prevalence among the HIV-negative women by facility in fact may suggest non-representativeness of the sampling, or the samples of women from these locales/neighborhoods were indeed representative and there is significant variability in the population risk within the Kweneng East District catchment area.

Second, we did not have histologic endpoints or surrogates of cancer risk to consider potential tradeoffs in cervical cancer risk reduction vs. overtreatment by using different triage strategies. We can infer from a seminal international study on the attribution of cervical cancer to different HPV types [33] on what the impact of triaging hrHPV-positive women with HPV genotyping might be. We however cannot compare those tradeoffs to that of using VIA alone or in combination with HPV genotyping. Finally, we did not have data on CD4 cell counts for the WLWH, which would have allowed us to look at its impact on hrHPV prevalence. CD4 cell counts among WLWH has been shown to be an independent predictor of hrHPV prevalence[16;23;24].

## Conclusions

We present some of the first data on hrHPV prevalence in WLWH and HIV-negative women living in Botswana. As seen in other populations, we observed a significant overall difference

in the hrHPV prevalence between WLWH and HIV-negative women. We also found that HPV16 was an important predictor of the appearance of an acetowhite lesion, a finding that should be verified in other studies.

## Acknowledgments

We sincerely thank the women of Kweneng East District in Botswana who participated in this research, without whom we would have not been able to conduct this study. We thank the research nurses: Clever Manyenyengwa, Thebeyame Diswai, Letang Gaofiwe, Thabiso Doreen Moiketsi, Omphemetse Mmunyane, and Rebecca Ketlametswe. We also thank the nurses and doctors who helped evaluate and treat the women who tested HPV positive: Olga Mokgatle, Mokgabo Queen Nonyane, Lesego Chigagane, Dr. Maduke Kula, Dr. Monica Malunga, Dr. Thinambo Mondali, and Dr. Rebecca Luckett. We further thank the following for their contributions to the design, implementation and data analysis of the study: Eva Bazant, Jennifer Snyder, Rosinah Dialwa, Tracey Shissler, Megan Wysong, Tebogo Kenosi, and Tlhomamo Pheto.

## Author Contributions

**Conceptualization:** Philip E. Castle, John E. Varallo, Margaret Mary Bertram, Bakgaki Ratshaa, Kereng Rammipi.

**Data curation:** Philip E. Castle, Margaret Mary Bertram.

**Formal analysis:** Philip E. Castle, John E. Varallo, Margaret Mary Bertram, Kereng Rammipi.

**Funding acquisition:** John E. Varallo.

**Investigation:** Philip E. Castle, John E. Varallo, Margaret Mary Bertram, Moses Kitheka, Kereng Rammipi.

**Methodology:** Philip E. Castle, John E. Varallo, Margaret Mary Bertram, Bakgaki Ratshaa, Kereng Rammipi.

**Project administration:** John E. Varallo, Margaret Mary Bertram, Bakgaki Ratshaa, Kereng Rammipi.

**Resources:** John E. Varallo, Margaret Mary Bertram, Bakgaki Ratshaa, Moses Kitheka, Kereng Rammipi.

**Supervision:** John E. Varallo, Margaret Mary Bertram, Bakgaki Ratshaa, Moses Kitheka, Kereng Rammipi.

**Validation:** John E. Varallo, Margaret Mary Bertram, Kereng Rammipi.

**Visualization:** John E. Varallo, Margaret Mary Bertram, Kereng Rammipi.

**Writing – original draft:** Philip E. Castle, John E. Varallo.

**Writing – review & editing:** Philip E. Castle, John E. Varallo, Margaret Mary Bertram, Bakgaki Ratshaa, Moses Kitheka, Kereng Rammipi.

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
