## [Decision Letter · Decision Letter 0]

21 Jan 2020

PONE-D-19-35746

High-risk human papillomavirus prevalence in self-collected cervicovaginal specimens from human immunodeficiency virus (HIV)-negative women and women living with HIV living in Botswana

PLOS ONE

Dear Dr Varallo,

Thank you for submitting your manuscript to PLOS ONE. After careful consideration, we feel that it has merit but does not fully meet PLOS ONE’s publication criteria as it currently stands. Therefore, we invite you to submit a revised version of the manuscript that addresses the points raised during the review process.

We would appreciate receiving your revised manuscript by Mar 06 2020 11:59PM. To enhance the reproducibility of your results, we recommend that if applicable you deposit your laboratory protocols in protocols.io, where a protocol can be assigned its own identifier (DOI) such that it can be cited independently in the future. For instructions see: http://journals.plos.org/plosone/s/submission-guidelines#loc-laboratory-protocols

We look forward to receiving your revised manuscript.

Kind regards,

Linus Chuang

Academic Editor

PLOS ONE

Journal Requirements:

3. Thank you for including your ethics statement:  "Johns Hopkins School of Public Health (JHSPH) Institutional Review Board IRB00007974".   

Please amend your current ethics statement to confirm that your named institutional review board or ethics committee specifically approved this study.

Philip E. Castle has received discounted or free HPV tests and assays for research purposes from Roche, Bectin Dickinson, Cepheid, and Arbor Vita Corporation. Otherwise, we have no conflicts of interest to disclose.

Reviewers' comments:

Reviewer's Responses to Questions

**Comments to the Author**

1. Is the manuscript technically sound, and do the data support the conclusions?

Reviewer #1: Yes

Reviewer #2: Yes

2. Has the statistical analysis been performed appropriately and rigorously? 

Reviewer #1: Yes

Reviewer #2: Yes

3. Have the authors made all data underlying the findings in their manuscript fully available?

Reviewer #1: Yes

Reviewer #2: Yes

4. Is the manuscript presented in an intelligible fashion and written in standard English?

Reviewer #1: Yes

Reviewer #2: Yes

5. Review Comments to the Author

Reviewer #1: This research article on hrHPV prevalence among HIV negative and WLWH in Botswana based on self-collected sample is a pilot demonstration study that documents success of self-collection in the Botswana context and provides new and insightful data on association of hrHPV and HIV. This is an impactful study and significant to the field of HIV/HPV and advances science on the use of HPV based screening in limited resource setting both among HIV negative and WLWH.

1. Methods (population)- Recruitment at various facility based on site (health facility vs. community) is difficult to follow based on text presented. It is unclear if two of the recruitment sites only had community health based outreach recruitment?

2. Result (Table 4)- Data as presented in the text does not match table 4. Furthermore, the numbers as presented as total visible lesion do not add up. Please clarify

Reviewer #2: The manuscript addresses a current topic as self collection is an important way to increase screening coverage if HPV testing is available in a low resource setting . Overall the manuscript was well written and worthy of publication. Some minor comments noted:

Introduction should reference that self collections was previously found to be acceptable to women in Botswana (ex reference previous studies on acceptability). .

Page 9 Methods: last paragraph beginning with "The outcome of VAT.......HPV16+ ELSE, HPV negative........".

Does ELSE imply versus: please clarify

Page 12 Results section: paragraph following Table 2 that begins with "Interestingly, the hrHPV prevalence differed between......"

This was difficult to follow since # clinic vs # community referrals were not identified for any of the clinics (A,B, C, D, or E). Therefore it was difficult to assess the prevalence data given, without knowing n values.

6. PLOS authors have the option to publish the peer review history of their article (what does this mean?). If published, this will include your full peer review and any attached files.

Reviewer #1: No

Reviewer #2: No

---

## [Author Response · Author response to Decision Letter 0]

28 Jan 2020

Reviewers' comments:

Reviewer's Responses to Questions

Comments to the Author

1. Is the manuscript technically sound, and do the data support the conclusions?

Reviewer #1: Yes

Reviewer #2: Yes

2. Has the statistical analysis been performed appropriately and rigorously? 

Reviewer #1: Yes

Reviewer #2: Yes

3. Have the authors made all data underlying the findings in their manuscript fully available?

Reviewer #1: Yes

Reviewer #2: Yes

4. Is the manuscript presented in an intelligible fashion and written in standard English?

Reviewer #1: Yes

Reviewer #2: Yes

5. Review Comments to the Author

Thank you very much for the insightful and useful feedback to help make this a stronger manuscript. Please see below my responses.

Reviewer #1: This research article on hrHPV prevalence among HIV negative and WLWH in Botswana based on self-collected sample is a pilot demonstration study that documents success of self-collection in the Botswana context and provides new and insightful data on association of hrHPV and HIV. This is an impactful study and significant to the field of HIV/HPV and advances science on the use of HPV based screening in limited resource setting both among HIV negative and WLWH.

1. Methods (population)- Recruitment at various facility based on site (health facility vs. community) is difficult to follow based on text presented. It is unclear if two of the recruitment sites only had community health based outreach recruitment?

Thank you for pointing that out. I hope the following is clearer.

All five health facilities conducted recruitment at their respective facilities, while two health facilities (B and E) also conducted recruitment in the community during health outreach activities. In these two facilities, recruitment was planned to be half and half from the health facility and in the community each serves through existing integrated community health outreaches, but 70% were recruited by community outreach for one facility (B).

2. Result (Table 4)- Data as presented in the text does not match table 4. Furthermore, the numbers as presented as total visible lesion do not add up. Please clarify

Thank you very much for catching that. The text was correct. It has been reviewed and corrected in the table, which is now Table 5, since we added tables to capture ‘data not shown’.

Reviewer #2: The manuscript addresses a current topic as self collection is an important way to increase screening coverage if HPV testing is available in a low resource setting . Overall the manuscript was well written and worthy of publication. Some minor comments noted:

Introduction should reference that self collections was previously found to be acceptable to women in Botswana (ex reference previous studies on acceptability). 

This is an excellent point regarding acceptability of self-collection, in other settings. At the time of our study, however, data on acceptability in Botswana was not available. However, our study did collect that information and we are in the process of writing a separate paper on the feasibility and acceptability aspects of the study.

Page 9 Methods: last paragraph beginning with "The outcome of VAT.......HPV16+ ELSE, HPV negative........".

Does ELSE imply versus: please clarify

Agree – versus is more clear and this has been changed.

Page 12 Results section: paragraph following Table 2 that begins with "Interestingly, the hrHPV prevalence differed between......"

This was difficult to follow since # clinic vs # community referrals were not identified for any of the clinics (A,B, C, D, or E). Therefore it was difficult to assess the prevalence data given, without knowing n values.

This was addressed through the above comment, and copied again below.

All five health facilities conducted recruitment at their respective facilities, while two health facilities (B and E) also conducted recruitment in the community during health outreach activities. In these two facilities, recruitment was planned to be half and half from the health facility and in the community each serves through existing integrated community health outreaches, but 70% were recruited by community outreach for one facility (B).

6. PLOS authors have the option to publish the peer review history of their article (what does this mean?). If published, this will include your full peer review and any attached files.

Do you want your identity to be public for this peer review? For information about this choice, including consent withdrawal, please see our Privacy Policy.

Reviewer #1: No

Reviewer #2: No

---

## [Editor Report · Decision Letter 1]

30 Jan 2020

High-risk human papillomavirus prevalence in self-collected cervicovaginal specimens from human immunodeficiency virus (HIV)-negative women and women living with HIV living in Botswana

PONE-D-19-35746R1

Dear Dr. Varallo,

We are pleased to inform you that your manuscript has been judged scientifically suitable for publication and will be formally accepted for publication once it complies with all outstanding technical requirements.

With kind regards,

Linus Chuang

Academic Editor

PLOS ONE
---

## [Editor Report · Acceptance letter]

7 Feb 2020

PONE-D-19-35746R1 

High-risk human papillomavirus prevalence in self-collected cervicovaginal specimens from human immunodeficiency virus (HIV)-negative women and women living with HIV living in Botswana 

Dear Dr. Varallo:

I am pleased to inform you that your manuscript has been deemed suitable for publication in PLOS ONE. Congratulations! Your manuscript is now with our production department. 

With kind regards,

on behalf of

Dr. Linus Chuang 

Academic Editor

PLOS ONE